# Microporous Hydroxyapatite-Based Ceramics Alter the Physiology of Endothelial Cells through Physical and Chemical Cues

**DOI:** 10.3390/jfb14090460

**Published:** 2023-09-05

**Authors:** Julie Usseglio, Adeline Dumur, Esther Pagès, Émeline Renaudie, Alice Abélanet, Joël Brie, Éric Champion, Amandine Magnaudeix

**Affiliations:** Université de Limoges, CNRS, Institut de Recherche sur les Céramiques, UMR 7315, F-87000 Limoges, France; julie.usseglio-grosso@unilim.fr (J.U.); adeline.dumur@etu.unilim.fr (A.D.); esther.pages@unilim.fr (E.P.); emeline.renaudie@unilim.fr (É.R.); alice.grenier-abelanet@unilim.fr (A.A.); joel.brie@unilim.fr (J.B.); eric.champion@unilim.fr (É.C.)

**Keywords:** hydroxyapatite ceramics, chemical substitution, microporosity, endothelial cells, vascularization, tubulogenesis

## Abstract

Incorporation of silicate ions in calcium phosphate ceramics (CPC) and modification of their multiscale architecture are two strategies for improving the vascularization of scaffolds for bone regenerative medicine. The response of endothelial cells, actors for vascularization, to the chemical and physical cues of biomaterial surfaces is little documented, although essential. We aimed to characterize in vitro the response of an endothelial cell line, C166, cultivated on the surface CPCs varying either in terms of their chemistry (pure versus silicon-doped HA) or their microstructure (dense versus microporous). Adhesion, metabolic activity, and proliferation were significantly altered on microporous ceramics, but the secretion of the pro-angiogenic VEGF-A increased from 262 to 386 pg/mL on porous compared to dense silicon-doped HA ceramics after 168 h. A tubulogenesis assay was set up directly on the ceramics. Two configurations were designed for discriminating the influence of the chemistry from that of the surface physical properties. The formation of tubule-like structures was qualitatively more frequent on dense ceramics. Microporous ceramics induced calcium depletion in the culture medium (from 2 down to 0.5 mmol/L), which is deleterious for C166. Importantly, this effect might be associated with the in vitro static cell culture. No influence of silicon doping of HA on C166 behavior was detected.

## 1. Introduction

Biomaterials are used daily in medicine to reconstruct bone losses. Hydroxyapatite-based ceramics (HA-Ca_10_(PO_4_)_6_(OH)_2_) are biocompatible and osteoconductive. They have been widely implanted as porous scaffolds in bone regenerative medicine since the 1980s [1,2], but an improvement of their ability to be vascularized is required to extend their application.

Indeed, osteointegration of the scaffold is the most important condition for its long-term tolerance by living tissues. A lack of vascularization limits the in-depth colonization of scaffolds by newly formed tissues. Vascular endothelial cells play an active role in tissue repair through molecular communication with resident cells in the injured tissue [3,4,5]. Because cells are sensitive to the chemical–physical properties of their environment, tuning the chemical and/or micro-architectural designs of the scaffolds may improve endothelial cell growth at the ceramic surface [6,7].

For the first strategy, which consists of tuning the chemical design, of the various species to have been studied, silicon has been described as acting positively on endothelial and bone-forming cell activity [8,9,10,11]. Therefore, silicon-doped hydroxyapatite (SiHA) Ca_10_(PO_4_)_6−x_(SiO_4_)_x_(OH)_2−x_ has been widely investigated [12]. For example, in an in vivo study in sheep, SiHA ceramics were employed in association with the vascular endothelial growth factor, VEGF, in order to increase their biological properties, notably with respect to vascularization [13]. However, this chemical substitution in the HA lattice changes not only the chemical properties of the ceramic, but also its mechanical and physico-chemical properties (grain size, surface energy, wettability, solubility, etc.) [7,14]. Any of these parameters may also influence the biological properties [15]. In the case of silicon, in a review published in 2009, M. Bohner noted a positive effect of silicon substitution on the biological behavior, but without evidence of a direct action of silicon [16]. Silicon was also found to stimulate angiogenesis in vitro and in vivo [8,9,17].

A second strategy is to modify the microstructural design of the scaffold in order to change the biomaterial–biological fluid interface. For example, the presence of open microporosity (i.e., pore sizes of about 1 to 10 µm), by increasing the specific surface area, leads to the modification of several properties, notably the solubility of the ceramic, and consequently the release rate of chemical species into the surrounding fluids [18]. Similarly, surface microporosity increases protein absorption from the biological fluids, which facilitates the adhesion of bone cells [19]. Microporosity also has a positive effect on osteointegration, and it has been found to be associated with an enhanced vascularization of bioceramics in vivo [20]. The introduction of microporosity also affects surface topography and roughness, which are known to modulate cell behavior [21,22,23], but not to the same extent or in same way depending on the cell type [24]. Thus, not only osteoblastic lineages, but also cells from other systems and tissues, and more particularly endothelial cells, have to be considered in order to design optimized bioceramics. Moreover, endothelial cells are particularly sensitive to mechanical cues [6].

To the best of our knowledge, very few data are available about direct relationships between biomaterial chemical–physical parameters and endothelial cells. Two studies have shown in vitro that endothelial cells and endothelial precursor cells are sensitive but respond differently to titanium surface roughness [25,26].

Currently, the ability of bioceramics to be vascularized has been extensively studied in vivo, but this does not allow specifically exploring the response of endothelial cells to the biomaterial surface (e.g., [13,27]). There are protocols for assessing the influence of ceramic scaffold properties on its vascularization ability ex ovo [28], but very few in vitro. These in vitro methods, common for studying endothelial cell biology, are mostly applied indirectly and do not take account of the material surface properties. Indeed, they are carried out on dissolved extracts of ceramics or on dense materials [9,29,30]. It is then difficult to draw conclusions regarding the impact of the physical parameters of ceramics on endothelial cells in vitro, making it difficult to tune the architectural design in order to enhance endothelial cell activity. Angiogenesis is the growth of new blood vessels from pre-existing ones. Its characterization directly on/in the biomaterial is therefore essential. Tubulogenesis is performed by activated endothelial cells following a pro-angiogenic stimulus in a 3D environment [31], and it recapitulates events that occur during angiogenesis at the endothelial cell level. It corresponds to the formation of tubules (assembly of cells around a central lumen) that will organize themselves into a network [32,33].

The present study was performed in order to understand the biological response of the murine C166 embryonic endothelial cells, which are able to perform physiological tubulogenesis, at the surface of four HA-based ceramics [34]. The HA ceramics varied in terms of their chemical and physical parameters: pure HA, Ca_10_(PO_4_)_6_(OH)_2_ [35], or silicon-doped HA (SiHA), Ca_10_(PO_4_)_5.6_(SiO_4_)_0.4_(OH)_1.6_ [36], in the form of either dense or microporous (25% open microporosity) parts. Dense HA (dHA) ceramics were considered as the reference control, as they have been used extensively in clinical applications for more than 40 years. Metabolic activity, adhesion and proliferation were studied as well as the expression of protein associated with the endothelial cells’ angiogenic status.

An experimental method was developed to assess tubulogenesis in vitro that allowed the discrimination of the influence of chemical composition from that of the micro-architectural design of the scaffold. To this end, a protocol using a fibrin-based gel layer deposited on the ceramic surface was set up. Two different assays were carried out: (i) a configuration where the cells were seeded directly in contact with the bioceramic surface (under the fibrin gel) to study the influence of the chemical and micro-architectural designs of the ceramic surface; and (ii) a configuration in which the cells were seeded on the fibrin gel to study the influence of dissolution products of the ceramic in culture medium.

The four HA-based ceramics used in this study allowed us to study the accuracy of the method and evaluate the influence of the chemistry and the microstructure on the endothelial cells. Dissolution products were determined, and their effect on the cells was also assessed by an indirect wound healing assay.

## 2. Materials and Methods

### 2.1. Ceramic Processing and Characterization

HA (Ca_10_(PO_4_)_6_(OH)_2_) and SiHA (Si molar ratio: 0.4; Ca_10_(PO_4_)_5.6_(SiHO_4_)_0.4_(OH)_1.6_) powders were synthetized by aqueous precipitation. Detailed protocols can be found in previous works [35,37]. The raw powders thus obtained were calcined at 650 °C for 30 min for HA and 700 °C for 2 h for SiHA, under air atmosphere (LH30/13, Nabertherm, Lilienthal, Germany), to reach a specific surface area of around 30 m^2^/g, which facilitated the compaction of the powders. The specific surface area of the powders was checked according to the eight-point Brunauer-Emmett-Teller method (BET-Micromeritics ASAP 2000, Micromeritics, Norcross, GA, USA) using adsorption of N_2_ after degassing under vacuum at 200 °C for 1 h. Afterwards, pellets were shaped by uniaxial pressing at 125 MPa in a 10 mm diameter cylindrical die (Specac, London, UK).

The sintering cycles were adjusted to obtain a similar microstructure for dense HA and SiHA ceramics (referred as dHA and dSiHA), on the one hand, and for porous ones (referred as pHA and pSiHA), on the other hand. The sintering parameters were 1200 °C for 30 min for dHA and 1240 °C for 30 min for dSiHA (LHT04/17 furnace, Nabertherm, Lilienthal, Germany), and 1000 °C for 30 min and 1160 °C for 30 min (LH30/13 furnace, Nabertherm, Lilienthal, Germany) for pHA and pSiHA, respectively. Sintering was performed under air atmosphere.

The phase purity of the sintered ceramics was checked by X-ray diffraction (XRD) using CuKα radiation (D8 Advance, Bruker, Saarbrücken, Germany). Crystalline phase identification was performed by comparing the experimental diagrams with the powder diffraction files (PDF) of the International Centre for Diffraction Data (ICDD). The chemical nature of ionic groups was controlled by Fourier-transform infrared spectroscopy (FTIR) using a Perkin Elmer Spectrum One spectrometer (Perkin Elmer, Shelton, CT, USA). Powdered samples of sintered pellets were mixed with KBr, and the absorbance spectra were recorded over the 4000–400 cm^−1^ region for 32 scans with a resolution of 4 cm^−1^.

Open porosity of pellets (P, expressed as mean ± standard deviation (SD) of measurements made on eight pellets) was determined using Archimedes’ method by weighing dried, wet and immersed pellets in ultrapure water. The specific surface area of pellets was determined by the BET method (Micromeritics ASAP 2000) using adsorption of Kr (for dHA and dSiHA) or N_2_ (for pHA and pSiHA) after degassing pellets under vacuum at 200 °C for 1 h. The microstructure of the ceramics was examined by scanning electron microscopy (SEM) using a Quanta 450 FEG (FEI, Hillsboro, OR, USA). The grain size distribution and average grain size (Φ_G_) of pellets were determined by SEM image analysis. Pore size distribution and average pore size (Φ_P_) were also determined by SEM image analysis. Prior to in vitro assays, the pellets were sterilized in an oven that delivered a dry air heat of 200 °C for 2 h (Memmert GmbH, Schwabach, Germany).

### 2.2. In Vitro Assays

#### 2.2.1. Cell Culture

C166 (CRL-2581, ATCC), immortalized endothelial cells of murine origin were routinely cultured in high-glucose (4.5 g/L D-glucose) Dulbecco’s Modified Eagle Medium (DMEM, Gibco, New York, NY, USA), supplemented by 10% fetal calf serum (FCS) (ThermoFisher, Carlsbad, CA, USA), penicillin (50 UI·mL^−1^) and streptomycin (50 µg·mL^−1^) (Gibco—complete culture medium), at 37 °C, under 5% CO_2_ wet atmosphere. The medium was changed every 3 days and the cells split when the confluence reached 60 at 80%.

In order to establish a chemical equilibrium between the ceramic surface and the culture medium, the samples were immersed in complete culture medium for 1 h at 37 °C, which was discarded before cell seeding. Cell seeding was performed dropwise on the top of ceramic pellets placed into 24-well culture plates (Hildesheim, Germany) at different cell densities depending on the experimental requirements. When experimental culture was continued for 7 days, the culture medium was renewed after 3 days.

#### 2.2.2. Tubulogenesis

Fibrin-based gels were prepared by mixing 300 µL of bovine fibrinogen (MP Biomedicals, Irvine, CA, USA) solution at 6 mg/mL in PBS 1X (Gibco) with 300 µL of high-glucose DMEM in the culture wells. Then, 300 µL of a 25 UI/mL thrombin solution (Merck, Rahway, NJ, USA) in pure H_2_O was added for polymerization. The fibrin-based gel was left to solidify for 45 min at 37 °C.

In configuration (i), the cells were directly seeded on the ceramic pellet at 20,000 cells/cm^2^ and incubated for 1 h. Afterwards, the pellets were transferred to another well, and the fibrin gel was layered onto the cell-seeded ceramic surface. After solidification, 1 mL DMEM was added in the well for cell cultivation. In configuration (ii), fibrin-based gels were directly poured on the pellets. After, solidification, the C166 cells were seeded on the gel at 20,000 cells/cm^2^ in 1 mL of complete culture medium.

The cells were cultured for 7 days, and the culture medium was renewed after 3 days. After 7 days, cells were fixed with 4% PFA for 10 min at RT.

#### 2.2.3. Viability, Cell Density and Proliferation

C166 were seeded at a density of 2500 cells/cm^2^ on the 4 ceramic conditions. Four hours before the end of the experiment, a volume equivalent to 10% *v*/*v* of the culture well volume of 0.2 mg/mL solution of resazurin (Acros Organics, Geel, Belgium, Verona, Italy) freshly prepared PBS 1× (Gibco) was added, and the cells were incubated at 37 °C. After 4 h, the culture supernatant was harvested, and fresh complete medium was added to the wells. Then, 100 µL of supernatant was transferred into black 96-well plates (Corning, New York, NY, USA) for fluorescence intensity measurement at 544 nm in a fluorimeter (FLUOstar OPTIMA, BMG Labtec, Offenburg, Germany). The cell viability assay was performed after 1, 2, 4 and 7 days of culture.

For density and proliferation assessments of the C166 cells, 20 µM of EdU (5-ethynyl-2′-deoxyuridine-, ThermoFisher, Carlsbad, CA, USA) was added to the culture wells and incubated for 1 h until the end of the experiment. Then, the cells were fixed in a 4% paraformaldehyde solution (PFA, Sigma Aldrich, St. Louis, MO, USA) for 15 min at RT (room temperature), rinsed 3 times in PBS 1× (Gibco), incubated in 100 mM Tris solution pH 7.6 (Acros Organics) for 10 min, and permeabilized in 0.1% *v*/*v* Triton-X100 (ThermoFisher) in PBS 1× for 15 min at RT. Cells were then incubated for 30 min at RT in the dark in a click-chemistry reaction buffer composed of 2 mM CuSO_4_ (Sigma Aldrich), 8.3 mM bright photostable AzideFluor^®^ 488 (Sigma Aldrich) and 20 mg/mL ascorbic acid (Acros Organics) subsequently added in 1× PBS. A counterstaining of nuclear DNA was carried out with 20 µM Hoescht 33342 in PBS 1× for 5 min at RT in the dark. At least 10 images were taken for each condition at X100 magnification with an epifluorescence microscope (AxioImager M2, Carl Zeiss, Jena, Germany) for further image analysis in order to determine cell density and proliferation rate.

#### 2.2.4. Dosage of VEGF-A Secretion by ELISA

C166 were seeded at 2500 cells/cm^2^ on the 4 materials. The negative control was C166 seeded without material (plastic of culture dishes). After 24 h, 72 h and 168 h of culture, the culture supernatant was harvested for VEGF-A dosage by ELISA (splicing variants VEGF_164_ and VEGF_120_). ELISA was performed on the culture supernatants according to the manufacturer’s instructions (DuoSet ELISA Development System, Mouse VEGF, R&D Systems, Minneapolis, MN, USA).

#### 2.2.5. Western Blot

After 7 days of culture, C166 cells were rinsed three times in PBS 1× and lysed in Laemmli buffer 1× without bromophenol blue and beta-mercaptoethanol (2% *w*/*v* sodium dodecyl sulfate (SDS) (Sigma-Aldrich), 10% *v*/*v* glycerol (Sigma-Aldrich), 60 mM Tris (ThermoFisher Scientific), pH 6.8). The lysates were then passed through a 28G syringe needle and centrifuged at 17,000× *g* for 10 min at RT. The protein content was dosed with a UV-vis spectrophotometer (QuickDrop, Molecular Devices, San Jose, CA, USA). Then, 5% beta-mercaptoethanol (Acros Organics) and 0.002% of bromophenol blue (Sigma Aldrich) were added to the lysates, which were then heated at 95 °C for 10 min, for protein denaturation. For each condition, 15 μg of heat-denaturated proteins were loaded on 4–15% SDS–polyacrylamide gels (Bio-rad, Hercules, CA, USA) and separated by electrophoresis. Then, the proteins were transferred on 0.45 μm polyvylidene fluoride membranes (Bio-rad). After the saturation of the specific antigenic sites with 5% skimmed milk (Régilait, Saint-Martin-Belle-Roche, France) in TBS-0.5% Tween-20 for 1 h at RT, the membranes were first hybridized with primary antibody overnight at 4 °C (rabbit anti-MMP9 or mouse anti-actin, Sigma-Aldrich), and then with the appropriate secondary antibody conjugated to HRP (goat anti-rabbit; goat anti-mouse, ThermoFisher) for 1 h at RT. The proteins of interest were detected by chemiluminescence using a luminol-based reagent (Amersham, Chaicago, IL, United States), the signal was detected and acquired using a Fusion FX system (Vilber-Lourmat, Marne-La-Vallée, France). The results were analyzed using the ImageJ software (version 1.54f, NIH, Bethesda, MD, USA) and the densities of the signals of interest were normalized according to those of actin. The same membranes were used to analyze MMP-9 and actin expression. A stripping procedure was performed after MMP-9 signal revelation by immersing membranes for 10 min at RT in a stripping buffer (2% *w*/*v* glycine (Sigma-Aldrich), 1% *w*/*v* SDS, 10% *w*/*v* Tween-20, pH 2.2). Then, the Western blot procedure was restarted from the saturation step.

#### 2.2.6. In Situ Expression of Angiogenesis Markers

Staining of the actin cytoskeleton or immunolabeling of angiogenesis-related proteins was performed either on C166 cells directly cultured at the surface of the ceramics (seeded at 20,000 cells/cm^2^ on the 4 materials and cultured for 7 days, then fixed with 4% PFA for 10 min at RT) or on cells used for the tubulogenesis assays. Though not systematically specified, the cells were rinsed three times in PBS 1× between each step of the procedure. The cells were permeabilized in PBS/Triton X-100 (ThermoFisher) 0.1% *v*/*v* for 15 min at RT. Aspecific antigenic sites were saturated with 3% m/v BSA (Bovine Serum Albumin, Sigma-Aldrich) in PBS 1× solution for 30 min at RT. Actin cytoskeleton was stained by incubation with phalloidin conjugated to a fluorochrome (Santa Cruz Biotechnology, Dallas, TX, USA) for 1 h at RT according to the manufacturer recommendations. VEGFR-2 and MMP-9 were labeled using monoclonal mouse-anti-VEGFR-2 (clone A-3, Santa Cruz Biotechnology, dilution 1:200) and polyclonal rabbit anti-MMP9 (Sigma Aldrich, dilution 1:250), respectively, for 2 h at RT.

Secondary antibody (goat anti-mouse conjugated to AlexaFluor 488 and goat anti-rabbit conjugated to AlexaFluor 594, ThermoFisher) was hybridized for 1 h at RT in the dark. Counterstaining of the nuclei was performed with Hoechst 33342 (ThermoFisher), 10 µM in PBS 1× for 5 min at RT in the dark.

Images were acquired using an epifluorescence microscope (AxioImager M2, Carl Zeiss, Jena, Germany). For tubulogenesis assay, the images were acquired with a AZ100 wide-field confocal macroscope (Nikon, Tokyo, Japan) coupled to a Andor DU-897 camera (Oxford Instruments, England, UK).

#### 2.2.7. Dosage of Chemical Elements in the Culture Medium

During cell culture experiments, the culture medium was collected after 3 days (medium renewal) and 7 days (end of experiments). Additionally, ceramic pellets were immersed in the complete culture medium in the same conditions but without cells. The medium was collected after 3 and 7 days, pooled by condition, and diluted in pure water in order to obtain a final volume of 10 mL. Then, 2% of nitric acid (VWR) was added. Calcium and silicon contents were dosed into the collected medium by inductively coupled plasma optical emission spectroscopy (ICP/OES, Optima DV 8300, Perkin Elmer, Shelton, CT, USA). Pellet surfaces without cells seeded were examined after 7 days of immersion by scanning electron microscopy (Quanta 450 FEG, FEI, Hillsboro, OR, USA).

#### 2.2.8. Wound Healing Assay in Ceramics Dissolution Media

C166 were seeded at 25,000 cells/cm^2^ in 96-well culture plates (Sarstedt, Germany) for 48 h to obtain a confluent cell layer. A wound was realized with a Wound Maker (Essen Bioscience, Ann Arbor, MI, USA) in the center of cell layers. Cells were rinsed in complete culture medium and 100 µL of a test medium was added. In a first set of experiments, the test medium consisted either of complete culture medium (as described in Section 2.2.1) or of conditioned culture medium resulting from the immersion of ceramic pellets alone (without seeded cells) for 3 days at 37 °C in a cell culture incubator. In a second set of experiments, the test medium was either complete culture medium or high-glucose DMEM without calcium (Gibco), supplemented with 10% FCS, penicillin and streptomycin, and different concentrations of CaCl_2_ (0.5, 0.75, 1 and 1.5 mM, Sigma-Aldrich). The cells were then cultured for 48 h in an Incucyte system (Sartorius, Göttingen, Germany) equipped with a camera taking two images per well each hour. The wound area was measured for all of the resulting pictures using ImageJ software, and the percentage of wound closure was calculated as follows:(1)$$100-\frac{100\times {wound\;area}_{t}}{{wound\;area}_{t=0}}$$

Wound closure speed was calculated by linear regression (slope) from the curve of the percentage of the wound versus time during the first 24 h of the experiment.

### 2.3. Statistical Analysis

A minimum of 3 independent in vitro experiments were conducted for each biological evaluation. Data were assayed for normality using the Shapiro–Wilk test using GraphPad Prism 9 (version 9.5.1, GraphPad Software, San Diego, CA, USA). According to the data set and normality test results, the Kruskal-Wallis test was followed by a Dunn’s post hoc test, or one-way or two-way analysis of variance (ANOVA) was performed and followed by Tukey’s multiple comparison test. Differences were considered significant when *p* < 0.05. All of the results with error bars are presented as mean ± standard deviation. The values given for the analysis of metabolic activity were normalized on the absorbance measured for dHA at 24 h. The results of image analysis obtained for the quantifications of the Western blots were normalized to the dHA ceramic as a reference before statistical evaluation.

## 3. Results

### 3.1. Characterization of Hydroxyapatite-Based Ceramics

The chemical–physical characterizations of sintered ceramic pellets are summarized in Figure 1. The XRD patterns (Figure 1C) displayed only peaks matching those of ICDD-PDF card n°9-432, assigned to the apatite crystalline phase. No other crystalline phase was detected. FTIR spectra (Figure 1B) confirmed the phase purity of the HA and SiHA ceramics with the vibrational bands of phosphate (at 474, 600, 963, 1040 and 1090 cm^−1^) and hydroxyl (at 3572 and 630 cm^−1^) groups for HA ceramics, and additional bands of low intensities in the domain 500 to 960 cm^−1^ (Figure 1B) that were assigned to silicate vibrations in the apatite lattice of SiHA ceramics [38]. The typical microstructures of the four ceramics are given in Figure 1D. dHA and dSiHA had very similar microstructures, with monomodal grain and pore size distributions. Average grain size was Φ_G_ = 0.55 µm for both dHA and dSiHA. In the same way, pHA and pSiHA also had similar microstructures, with monomodal grain and pore size distributions. Average grain and pore size values were Φ_G_ = 0.22 µm and Φ_P_ = 0.30 µm, respectively, for pHA, and Φ_G_ = 0.19 µm and Φ_P_ = 0.45 µm, respectively, for pSiHA. The total microporosity rate was 22% for pHA and 24% for pSiHA. The specific surface area was 0.003 m^2^/g and 0.009 m^2^/g for dHA and dSiHA, respectively, and it reached 4.0 m^2^/g and 2.0 m^2^/g for pHA an pSiHA, respectively. As both the chemistry and the microstructure of bioceramics may affect their biological behavior, it was of prime importance to be able to discriminate between their respective effects. Thus, sintering conditions for each ceramic were chosen in order to be able to evaluate (i) the influence of the microstructure, by comparing dense versus porous ceramics of the same composition, and (ii) the influence of the chemistry, by comparing HA versus SiHA ceramics having the same microstructure.

### 3.2. Ceramics’ Biocompatibility towards C166 Endothelial Cells

The metabolic activity of the C166 endothelial cells was first evaluated with time from 1 to 7 days (Figure 2A). Dense HA ceramics were used as a biocompatible control, and consequently, the results were normalized as a function of the absorbance measured for dHA at 24 h. From seeding until the seventh day (168 h) of culture, the overall metabolic activity increased, implying a growth of the cell population at the ceramic surface. More precisely, between 24 h and 48 h, the metabolic activity of the cell population growing at the ceramic surface increased significantly for all conditions, which was no more the case between 48 h and 96 h. Then, at 7 days, the metabolic activity was significantly higher than that at 96 h for the dense ceramics (dHA 96 h vs. dHA 168 h: *p* = 0.007, and dSiHA 96 h vs. dSiHA 168 h: *p* < 0.001), but not for the porous ones (pHA 96 h vs. pHA 168 h: *p* = 0.621, and pSiHA 96 h vs. pSiHA 168 h: *p* = 0.545). At this culture time, the metabolic activity was also significantly higher when cells were cultivated on the surface of dSiHA ceramics compared to that of other ceramic materials. While not statistically significant, a trend towards a lower metabolic activity on porous ceramics compared to dense ones could be observed. The metabolic activity was notably dependent on the number of cells. Therefore, the cell density at the surface of the material was evaluated by image analysis at 24 h and 7 days (Figure 2B). The cell density trended to be higher on dense ceramics compared to on porous ones. After 7 days, all of the cells were confluent. Whatever the time point, very few condensed nuclei or nuclei with altered morphologies were seen, suggesting a low cell mortality, comparable to that observed in routine cultures of C166 cells. The proliferation ability of the cells was evaluated at 24 h, before confluency (Figure 2C). At this time, the number of cells positive for EdU was significantly reduced (by around half) at the surface of the microporous samples compared to at the surface of their dense counterparts. There were no significant differences with respect to the chemical composition of the ceramics.

### 3.3. Expression of Angiogenesis Markers by C166 Cells Cultured on Ceramics

To assess the angiogenic status of the C166 cells cultivated onto the different ceramic surfaces, the secretion of the key angiogenic growth factor VEGF-A isoform 164 was dosed by ELISA at three time points: 24 h, 72 h and 168 h (Figure 3A). At 24 h, about 50 pg/mL of VEGF-A was detected in the culture supernatant of the C166 cells, regardless of the culture support (ceramics or culture plastic). After 72 h, there were no significant difference between the conditions, while a trend towards higher secretion of VEGF-A by C166 cells cultured onto ceramic surfaces compared to C166 cells cultured in the bottom of plastic culture dishes was observed. This trend was confirmed after 7 days of culture (168 h) with a significantly higher concentration of VEGF-A detected in the culture supernatant of C166 cells grown at the surface of ceramics than in that of cells grown directly on plastic. In all of the experiments, neither the chemical composition nor the microstructure of the ceramic pellets influenced VEGF-A secretion.

To explore the physiological state of the C166 cells as a function of the ceramic surface onto which they were cultivated in more depth, MMP-9 expression in C166 was assayed by Western blot after 7 days of culture (Figure 3B). A major signal was detected at about 102 kDa corresponding to the pro-enzyme. The signal detected around 80 kDa should correspond to the active form of the enzyme resulting from a proteolytic cleavage. Whatever the considered form, pro- or active MMP-9, no significant differences were observed between any of the conditions (Figure 3B, lower panel).

The expressions of MMP-9 and the VEGF-A main receptor, VEGFR-2, were then studied in situ after immunostaining, at two magnifications (Figure 4 and Appendix A). On all ceramics, the cell population was homogeneously distributed throughout the available surface. MMP-9 expression was detected on all of the samples, but not all of the cells expressed this protein or expressed it with the same intensity. The staining was mostly intracytoplasmic in MMP-9-expressing cells (arrows, Figure 4 and Appendix A). The global intensity of the signal did not differ markedly between the conditions, which is consistent with the Western blot results (Figure 3B and Appendix A).

Regarding VEGFR-2 (Figure 4, Appendix A), the staining was difficult to analyze on porous ceramics because of a reflection phenomenon, due to the microscopy modality (epifluorescence), which produced a high background (examples of zones of high background are highlighted with stars on Figure 4 and Appendix A). VEGFR-2 was detected in the cytoplasm of all of the cells (Figure 4 and Appendix A), with no differences being observed, regard less of the chemistry or architecture of the ceramics.

Taken together, these results highlight that the ceramics exert a positive influence on the secretion of VEGF-A in comparison with the plastic dishes. This secretion was not altered by either the ceramic chemical composition or the surface microstructure. The analysis of angiogenesis markers downstream of VEGF-induced angiogenic pathways, VEGFR-2 and MMP9 confirmed that the changes in ceramic microstructure or composition did not influence the angiogenic status of the C166 cells.

### 3.4. Tubulogenesis

In vitro tubulogenesis recapitulates the events occurring during angiogenesis at the endothelial cell level, when they are cultivated in a 3D environment mimicking an ECM, generally obtained using basement membrane-like hydrogels (e.g., Matrigel, collagen matrix, etc.). Therefore, to explore the potential angiogenic behavior of the C166 endothelial cells according to the HA-based ceramic composition or microstructure, a tubulogenesis assay adapted to the use of ceramic scaffolds was developed. To this end, two experimental setups (depicted in the diagrams on the upper part of Figure 5) were implemented in order to discriminate the influence of the microstructure from that of the chemical elements, in this case, the incorporation of silicon into the HA lattice of the SiHA ceramics. In the first setup, named “Configuration 1”, the cells were seeded directly onto the ceramic pellet surfaces, and a fibrin gel was generated on top. Configuration 1 was designed to make it possible to study the influence of the microstructure and chemical composition of the ceramic on the cell behavior. The second setup, called “Configuration 2”, was designed to assess the potential effects of chemical elements that are released from the ceramics (dissolution products) into the culture medium. To this end, the fibrin gel was poured directly onto the ceramic surface, and the cells were seeded on top, so that they were not in contact with the material surface.

After 7 days of culture in Configuration 1 (Figure 5A), some tubule-like structures were present, mainly on the dense ceramics. These tubule-like structures were positive for VEGFR-2 and MMP-9, which was consistent with the occurrence of tubulogenesis. Some cells with a “tip-cell”-like structure were observed (Appendix A). The whole cell tissue organization differed depending on the ceramics’ microstructures, but not on their chemical compositions. On the one hand, the tubule-like network appeared more developed on dense ceramics. On the other hand, the cells that were not involved in the tubule-like structures made a layer with an alveolar pattern at the surface of microporous ceramics. There was no perceptible effect due to the presence of silicon in SiHA, with the cell repartition being similar for comparable microstructures (i.e., dHA vs. dSiHA or pHA vs. pSiHA). These results indicate a strong influence of the microstructure upon the organization of the C166 endothelial cells and their ability to perform tubulogenesis at the ceramic surface.

Unexpectedly, the results obtained with Configuration 2 (Figure 5B) were very close to those described for Configuration 1. On dense ceramics, tubule-like structures were more frequently seen than on the porous ones. At the surface of pHA and pSiHA, the cell layer acquired an alveolar pattern. Consequently, the mechanism by which the ceramics affected the cell behavior stemmed from the surface physical parameters associated with the microstructure, but the final cue is chemical in nature.

### 3.5. Dissolution of HA-Based Ceramics

In order to examine the hypothesis arising from the results of the tubulogenesis assay, the elementary chemical compositions of the dissolution products of the ceramics were evaluated. Ca and Si were dosed by ICP-OES into the complete culture medium in which the ceramic pellets were immersed in the presence of or without C166 endothelial cells (Figure 6). To reproduce the cell culture conditions, the culture medium was renewed after 3 days at 37 °C.

DMEM high-glucose is formulated with a calcium concentration of 1.8 mmol/L. The addition of 10% of FCS, characterized by a calcium concentration ranging between 2.1 and 2.6 mmol/L (physiological bovine calcemia), led theoretically to a final calcium concentration in the complete culture medium of around 2 mmol/L. The dosage confirmed this value, with a calcium concentration of 2.04 mmol/L measured in the complete culture medium alone. This is represented by the green dotted line in Figure 6A. The results differed depending on the ceramic microstructure. For dense materials (dHA and dSiHA), the deviation from the complete culture medium, taken as a reference, was very small. Only a slight decrease in Ca concentration was measured. It always remained above 1.8 mmol/L, whether cells were present or not. The results did not change after culture medium renewal suggesting the establishment of a chemical equilibrium.

In the case of the porous ceramics (pHA and pSiHA), the calcium concentration in the medium in which the ceramics were immersed was drastically diminished (by at least half) in comparison with their dense counterparts or the complete culture medium. Phosphorus dosage, represented by the presence of orthophosphate ions in the culture medium, returned similar trends regarding the difference between dense and microporous ceramics after 3 days of immersion (Appendix A), with a decrease in phosphorus concentration when microporous ceramics were used. The decrease in calcium and phosphate ion concentrations in the liquid medium can be explained by the adsorption of these ions at the ceramic surface in a poorly organized non-apatitic hydrated layer, as described elsewhere [39]. This phenomenon was all the more important when using porous samples, as their surface area was much greater than that of the dense samples. With the zeta potential of the HA-based ceramic surface being negative, ranging from −21 mV to −25 mV (determined on the basis of powder from crushed sintered pellets using a Zetasizer (Malvern Panalytical)), a preferred adsorption of Ca^2+^ ions compared to phosphate ions could logically be expected. Such a phenomenon was not detectable by scanning electron microscopy (Appendix A), as it did not lead to the formation of new biological apatitic crystals at the surface of the ceramic as a result of the dissolution/precipitation process (with a solubility constant Ks ≈ 10^−117^ for Ca_10_(PO_4_)_6_(OH)_2_ at 37 °C [40]), as is usually observed when, for instance, HA ceramics are immersed in simulated body fluids.

Neither the complete culture medium nor the pure HA ceramics contained silicon. Thus, the silicon released from the ceramics immersed into the culture medium was only dosed for dense and porous silicon-doped HA conditions (Figure 6B). As silicon is not expected to precipitate in the biological apatites that could form from the dissolution/precipitation of HA-based ceramics in solution, the results after 7 days are reported as cumulative release (i.e., the silicon content measured in the culture medium after 3 days is added to that measured in the renewed medium after 7 days). A continuous release of silicon into the culture medium, with or without cells, was measured over time. The silicon quantity released by the porous SiHA ceramics was at least 25 times greater than that released by dense ceramics without cells. This result can also be explained by the increase in specific surface area of the ceramic due to microporosity. With cells, this difference in silicon release between dense and porous ceramics persisted, but it was reduced to around 12 times.

Taken together, these results show that the presence of microporosity caused a depletion of the calcium (and, to a lesser extent, of phosphorus) in the culture medium, regardless of the ceramic composition, and an important increase in silicon concentration for SiHA ceramics, likely due to the higher specific surface area of the microporous samples.

### 3.6. Effect of Ceramic Extracts on C166 Activation

In order to assess the effect on C166 cell behavior of the chemical changes in the culture medium due to the microporosity of the ceramics, an indirect wound healing assay using the dissolution products resulting from the immersion of the ceramic pellets was realized (Figure 7). A wound was created on the confluent cell layer. Its recolonization was recorded for 48 h, as all of the wounds were closed after 48 h of incubation. It could be observed visually (Figure 7A) that the cells cultured in the dissolution products from dense SiHA ceramics closed the wound the fastest. The wound closures were the slowest with the dissolution products from microporous ceramics (pHA and pSiHA). The observations were confirmed by measurements made using computer-assisted image analysis (Figure 7B,C). The Figure 7B depicts the wound closure according to time. During the first 24 h, the kinetics of the closure was almost linear. Then, the final phase of the closure proceeded at a slower rate. The average wound closure was above 99% after 36 h for cells cultured in culture medium conditioned by dSiHA, 41 h for dHA, 43 h for pSiHA and 46 h for pHA. The dSiHA condition presented the highest wound closure rate at all time points. This was statistically different from dHA at nine time points between 2 and 20 h (stars on Figure 7B). Conversely, pHA did not differ from pSiHA. Notably, it could be observed that an inflexion arose 6 h after the lesion with the microporous-ceramic-derived culture media, causing a shift between the curves corresponding to the culture medium from dense ceramics and those from porous ones. This time point corresponded to the beginning of the occurrence of visible effects due to microporosity of the ceramics. Figure 7C presents the slopes of the wound closure curves calculated for each condition during the first 24 h (linear part of the curves). The wound closure speed was significantly higher when the cells were cultured in the conditioned medium resulting from immersion of dense ceramics, and there were no significant differences between dHA and dSiHA. These results confirm that microporosity had a negative influence on C166 endothelial cell activity via a chemical-based mechanism.

### 3.7. Impact of Calcium Deficiency in the Culture Medium on C166 Activation

Calcium is an essential element for cell biology, and its ionic concentration in the culture medium was strongly altered by the presence of microporous ceramics. To test whether the calcium deficiency was the cause of the negative impact of the culture media conditioned by microporous ceramics on C166 activity, a complete culture medium containing calcium concentrations in the range of those measured by ICP-OES was also tested on cells in a wound healing assay experiment (Figure 8). The basal high-glucose DMEM routinely used for C166 culture contains 1.8 mmol/L of calcium as CaCl_2_. To reproduce the reduction of the calcium concentration, a high-glucose DMEM without calcium supplemented with 0.5, 0.75, 1 and 1.25 mmol/L of CaCl_2_ was used to prepare the complete culture media. As depicted in Figure 8A, the lower the calcium concentration, the slower the wound closure. Moreover, when there was less than 1.25 mmol/L of calcium in the basal medium (which corresponds to about 1.35 mmol/L of calcium in the complete culture medium), the cells did not close the wound after 48 h. These observations were confirmed by image quantification. Here, three independent experiments were performed with C166 at different passages: p2, p5 and p8, respectively. A similar trend was observed for all conditions, but the cells with the highest passage numbers seemed to better resist calcium deprivation (Figure 8C, where crosses represent each independent experiment, and Appendix A).

Due to the high standard deviation, only a significant difference at *p* ≤ 0.05 was observed between the standard complete culture medium (1.8 mmol/L of calcium) and the lowest calcium concentration of 0.5 mmol/L. Considering the wound closure speed, the cell migration and proliferation were significantly lowered for cells cultured with DMEM containing 0.5 mmol/L of calcium in comparison with those containing 1.0 mmol/L. The Ca concentrations of 0.75 and 1.0 mmol/L in basal medium (or, respectively, around 0.86 and 1.12 mmol/L, when taking account of the FCS supplementation) were very close to those measured for pHA and pSiHA (Figure 6A). It is worth noting that the healing rates measured with culture media conditioned by microporous ceramics and those in their counterparts in terms of calcium concentrations were highly comparable (Appendix A). Thus, the reduction in the amount of calcium in the solution is most likely at the origin of the impairment of endothelial cell activity caused by the microporosity of HA ceramics.

## 4. Discussion

The vascularization of scaffolds made of calcium phosphate ceramics is crucial for the success of their implantation and further osteointegration [6]. The vascularization extent and quality condition the penetration, survival and activity of bone tissue cells in porous scaffolds at depths beyond 1 cm from the surface [41]. Among the strategies able to stimulate the biological performances of ceramics, chemical design and the adjustment of the micro- and macro-architecture of the porous scaffold appear highly promising [6,24]. However, whatever the strategy followed, several chemical–physical parameters will inevitably be modified, making it difficult to elucidate the underlying mechanism affecting cell or tissue behavior [24]. In other words, it is still a challenge to associate a given chemical or physical parameter with a particular cell response. In addition, for a given parameter, such as surface topography, different cell types do not react in the same way. For example, osteoblasts and osteoclasts are not stimulated by the same level of roughness: at a micrometric scale, roughness is favorable for osteoblasts and deleterious for osteoclasts [42]. The specific relationships between the chemical–physical properties of bioceramics and the endothelial cells at the interface are still little explored. The prime motivation of this study was to develop an in vitro model that would make it possible to perform tubulogenesis assays on the ceramic surface in order to study endothelial cell colonization. In order to be efficient, the experimental procedure had to distinguish between the physical and chemical parameters at the material/cell interface in order to understand how changes in the chemical composition and/or the microstructure of the material can influence biological behavior. To validate the experimental setup, four different HA-based ceramics, varying either in terms of their chemical composition or in terms of their microstructure, were used to perform two-by-two comparison.

The second objective of this work was to evaluate (i) the influence of silicon as a doping element in HA and (ii) the influence of surface micropores on the biological response. While numerous works devoted to the in vitro characterization of such ceramics using bone or mesenchymal stromal cells have been published, controversy still exists as to the real role of each of these chemical or physical parameters. Moreover, as mentioned above, the specific relationships between HA-based ceramic properties and endothelial cells have been little explored to date.

Thus, to study chemical design, HA-based ceramics were compared to silicon-doped HA. Silicon was chosen because of its potential pro-angiogenic activity, which has been documented in the literature both in vitro and in vivo [7,11,27,43,44,45,46]. The majority of in vitro studies have been performed using dissolution products from silicon-containing materials such as calcium silicate [10,11,27]. Stimulation of the expression of VEGFR-2 was hence described in HUVEC cells exposed to calcium silicate extracts [44]. With the aim of assessing the impact of the microarchitecture, HA and SiHA ceramics were differentially sintered to obtain either dense materials or comparable microporous ones, in terms of chemical–physical surface characteristics. By increasing the specific surface area and modifying other ceramic surface characteristics such as roughness and topography, microporosity interferes with cell adhesion [47], proliferation [48,49] and fate [21].

In order to specifically answer the question about the response of endothelial cells to the bioceramic surface, a mono-culture model was necessary. Because of its ability to exhibit physiological endothelial cell behavior and to form tubules in 3D environments [34], the murine C166 endothelial cell line was chosen as the biological model. The C166 cells originated from the yolk sac—the first site of vasculogenesis and hematopoiesis during the embryonic development [50]—of mouse embryos issued from hypervascular transgenic mouse. The C166 clone has the ability to support embryonic hematopoiesis, unlike other clones derived from the same mouse embryos [34,51], suggesting specific properties close to those of endothelial precursor cells (EPC). Maes et al. used the term of hemangio-endothelial to describe the C166 cells [52]. This feature is interesting, because EPCs are cells present in the bone marrow, and are of great interest in the vascularization of engineered tissues and biomaterials [25,53,54] for their ability to perform vasculogenesis (de novo formation of blood vessels) and for their pro-angiogenic properties. Nevertheless, C166 are mostly regarded as vascular endothelial cells and used as such [55,56], e.g., in the context of biomaterial vascularization [57], with the advantages and the known biases of an established endothelial cell line [46].

The first step in our study consisted of checking the physiology of the C166 cells seeded at the surface of the four HA-based ceramics. At 24 h, the cell populations at the surface of the different materials were comparable. A constant increase in cell population was observed over time on all of the ceramic surfaces. The C166 cells presented a high proliferation rate overall, although it was lower on the microporous ceramic surfaces. Thus, all of the ceramic conditions can be considered to be biocompatible toward endothelial cells, which is consistent with the data in the literature [58,59]. From the results for cell density, proliferation and expression of angiogenesis-related markers, the most influential parameter with respect to cell response was the microstructure. The incorporation of silicon into the crystal HA lattice did not affect these cell properties. Considering that microporosity increased the surface roughness, this result appears consistent with those found with primary EPC cultured on the surface of titanium disks varying in terms of their surface properties, thus highlighting the stronger effect of roughness compared to other surface properties such as wettability [25]. In this study, the EPC exhibited a lower proliferation rate on rougher substrates. An et al. showed that human vascular endothelial cells (HUVEC) had a better viability and proliferation on smooth and hydrophilic titanium surfaces [26]. However, the orders of magnitude of the roughness of the titanium substrates (around 1.2 µm for the rougher ones) and the HA-based ceramics (average roughness Ra: 4.0 ± 0.3 and 8.1 ± 0.6 µm for dHA and dSiHA, respectively) were not the same in the present work. HA, while slightly hydrophobic, has a strong affinity to proteins in the culture medium, which encourages cell adhesion, meaning that this material is not strictly comparable with titanium. Several works conducted either in vitro with co-cultures [58], ex ovo [28], or in vivo [58] have shown calcium phosphate ceramics to have good biocompatibility with endothelial cells, but to the best of our knowledge, none has studied the impact of microporosity. Microporosity, by mechanically increasing the specific surface area, should enhance the amounts of proteins adsorbed from the culture medium in vitro. Microporosity has been shown to increase bone cell adhesion via this phenomenon [19]. Microporosity also enhanced the release of silicon in the culture medium, which reached around 800 mmol/L (22.73 mg/L) after 7 days without cells. The dosage method employed here did not allow the identification of the chemical form of the silicon released from the Si-doped HA ceramics. Silicon is incorporated as a silicate ion, so it should be released as silicate ions or as metabolizable silicic acid [60], as was modeled in a recent study by Chappell et al. [60]. Silicon is bioavailable as orthosilicic acid and has a positive role in bone metabolism [61] and bone mineralization [60], where its absence is deleterious for bone homeostasis. The daily dietary silicon intake is estimated to be from 20 to 50 mg [60,62]. No toxicity of dietary silicon was detected at a dose higher than 50,000 mg in rats [63]. Silicon is tightly regulated in the organism, and cell-specific transporters have recently been identified [64,65]. It has been shown that silicic acid produced by the degradation of silicon quantum dots is eliminated through urine [66]. In the context of the implantation of silicon-containing ceramics as bone graft substitutes, a potential toxicity due to silicon can be reasonably ruled out, especially as micro-/macroporous silicon-substituted calcium phosphate scaffolds are currently used in clinic [12,67,68].

We have previously demonstrated an excellent biocompatibility of 23% open microporosity in silicon-doped HA toward blood vessels in an ex ovo context [28]. Nevertheless, in the present study, there was no positive influence of ceramic surface microporosity on C166 cell attachment or adhesion at short culture times. A comparable density was observed at the end of the experiment for the different ceramics due to a continuous growth of the cells until confluence. This highly proliferative status is related to their nature as an established cell line, and should not be taken into account when judging their activation state. The initial proliferation rate (24 h post seeding) was, however, negatively affected by the presence of open porosity, and the overall metabolic activity of the cell population was also altered. With cell density being unmodified, the latter observation might be attributable to differences in cell activity resulting from the microstructure of the ceramic surfaces. The second step was then to study the angiogenic status of the C166 cells directly cultured on the ceramics. The C166 cells expressed angiogenesis-related markers such as VEGFR-2 or MMP-9 with no differences depending on the microstructure or chemistry of the ceramics on which they grew. Some differences were observed regarding the expression of the master angiogenic and vasculogenic growth factor VEGF-A. Interestingly, starting from 72 h, a trend towards a higher secretion of VEGF-A was detected on all of the ceramics in comparison with the culture dish polystyrene. The increase in VEGF-A secretion on ceramics became significant after 168 h, indicating a pro-angiogenic potential for calcium phosphate ceramics through an auto and paracrine effect, as observed by Chen et al. for different calcium phosphate ceramics, including HA [58]. Although there were no statistically significant differences between the test conditions, we noticed a trend towards a greater production of VEGF-A by cells growing on microporous ceramic surfaces. A parallel with the results obtained by Ziebart et al. with EPC on rough titanium surfaces is therefore seductive [25]. However, HUVEC exhibited a contrasting secretion behavior on the same titanium substrates [26]. Keeping in mind that the surfaces of titanium and HA ceramics are not similar, the modulation of VEGF secretion by C166 is more likely to be due to the nature of the substrate (HA-based ceramics). An explanation for the light incidence of microporosity on VEGF-A secretion by C166 cells may reside in the higher adsorptive capacity of microporous ceramics towards proteins present in the culture medium originating from FCS supplementation, thereby increasing the probability of interaction with cells. FCS, as an extracellular matrix from the blood of mammal fetuses in development, contains adhesive proteins and growth factors, among them pro-angiogenic ones. Incidentally, tubulogenesis can be influenced by the FCS batch for the same endothelial cells.

To explain how the microporosity affects the C166 physiology, the setup of our dual tubulogenesis assay at the bioceramic surface was particularly relevant. This assay enabled the formation of tubule-like structures, mainly on dense ceramics, while the C166 cell tissue rather took on an alveolar morphology on the microporous surfaces. The presence of some cells with a morphology close to that of tip cells suggests that the process is close to angiogenesis. However, vasculogenesis cannot be totally excluded, because in this in vitro context, no “pre-existing vessel” exists, and it is possible to reproduce vasculogenesis with endothelial cells in fibrin gels [69]. Here again, the most influential parameter was the microstructure. The pattern of cell repartition differed drastically, with the cell layer on the microporous ceramics having an “alveolar” pattern. On dense microstructures, the cell layer was more homogeneous and the tubule-like structures more frequent. Dense materials therefore seem to be more favorable to the tubulogenesis process. However, the “alveolar” pattern suggests cell rearrangement, and does not mean that cells were inactive when cultured on microporous ceramics. Very similar results were observed in both configurations devoted to distinguishing the impact of chemical parameters from those linked to physical characteristics. The first explanation is that the procedure was not efficient, and did not enable the initial aim to be completed. The second possibility is that the results from the tubulogenesis assay argue in favor of a “chemical effect” of the microstructure. To determine which was the case, a comparison of the conditioned media was performed that showed a dramatical decrease in calcium concentration with microporous ceramic substrates. The subsequent results from the indirect wound healing assays using dissolution media proved that the impact of microstructure on C166 endothelial cell behavior was definitely due to a chemical mechanism. Indeed, the wound healing assay, which reflected endothelial cell activation characterized by an elevated capacity to proliferate and migrate, demonstrated that the dissolution products from microporous substrates slowed down the wound closure. The presence of silicon appeared favorable to wound closure on dense materials (significant difference between dHA and dSiHA) but not on porous ones (pHA vs. pSiHA). This significant difference was not perceived when the closure speeds were compared. Therefore, under these conditions, it is not possible to conclude a stimulative effect of silicon on C166 endothelial cells.

The reproduction of calcium concentrations using additions of CaCl_2_ in the culture medium returned the same results. Consequently, the modification of calcium availability is most likely the causative element explaining how the microstructure affects the C166 cell physiology. Intracellular ionized calcium is a crucial second messenger for the accomplishment of angiogenesis driven by endothelial cells exposed to pro-angiogenic cues like VEGF-A [70]. The entry of Ca^2+^ into the vascular endothelial cells is also stimulated by angiogenic stimuli. Targeting calcium entry in endothelial cells is a strategy for preventing tumoral angiogenesis [71], and several transient calcium channels are involved in VEGF-induced angiogenesis [72]. Such a situation could not be reasonably imagined in vivo, due to the permanent flow of interstitial body fluids and blood circulation. Thus, it can be hypothesized that the results obtained in our study are consequences of the 2D static culture. This observation is particularly important, because 2D culture is a convenient and simple means for conducting preliminary experiments to characterize the biological properties of biomaterials, including bioactive ceramics. Despite medium renewal after 3 days, the calcium concentration remained low at the end of the experiment. Consequently, with such microporous bioceramics, the calcium concentration during in vitro experiments should be documented and should be taken into account before reaching conclusions.

## 5. Conclusions

This work aimed to (i) develop a reliable in vitro protocol for assessing the angiogenic status of endothelial cells directly on biomaterials through a tubulogenesis assay and (ii) to validate this protocol by studying the impact of microporosity and chemical composition of HA-based ceramics on C166 endothelial cells. To this end, four ceramics, differing in terms of their chemical composition (HA vs. SiHA) and their microstructure (dHA and dSiHA vs. pHA and pSiHA), were produced. Prior to performing the tubulogenesis assay, the adhesion and proliferation of C166 cells at the ceramic surface were evaluated over 7 days. The C166 cells adhered significantly better to dense surfaces than to microporous ones, and had a better proliferation rate after 24 h, while the latter should be taken with caution due to the immortalized nature of the C166 line. The metabolic activity of the C166 cells was significantly altered in the experiments performed with longer durations. The production of pro-angiogenic proteins by C166 cells was then assayed. The C166 cultivated at the surface of ceramics exhibited significantly higher pro-angiogenic growth factor VEGF-A production (up to double) compared to when grown at the plastic surface of conventional culture dishes. With time, VEGF-A production trended towards being higher (not significantly) for the cells at the surface of porous ceramics compared to the cells on dense ones. However, there was no statistically significant difference for other angiogenic proteins such as MMP-9 or VEGFR-2. The presence of silicon did not impact the studied cell parameters. Microporosity is consequently the major ceramic parameter affecting the C166 cells at this step. Considering the literature data, the impact of microporosity on C166 might be due to physical cues: increased roughness (adherence and proliferation) and increased specific surface area through enhancement of protein absorption (production of VEGF-A).

We developed a reliable qualitative procedure to directly evaluate tubulogenesis at the surface of biomaterials using a fibrin gel using two configurations according to the position of cells with regard to the ceramic surface. The underlying idea was to discriminate the effects occurring due to the ceramic’s chemistry from those originating from its chemical–physical parameters, mainly microporosity, in this case. Tubule-like structures were observed after 7 days, indicating that the assay was informative. On microporous ceramics, in both configurations, tubule-like structures were less frequently observed (qualitative observation); rather, a repartition of the cell layer into an alveolar pattern was observed.

Having been validated, this procedure will be upgraded by making it quantitative using adapted microscopy imaging and computed analysis and by using co-cultures to sustain endothelial cell assembly into capillary-like structures. The analysis of dissolution products from the ceramics and their impact on the ability of C166 cells to proliferate and migrate from a wound healing assay highlighted that the negative impact of microporosity on C166 is of chemical origin, and is due to a reduction in soluble calcium in cell’s surrounding fluids (culture medium). This was validated by modeling this phenomenon by varying the calcium concentration in the culture medium.

This work, conducted with an endothelial cell line alone, aimed to demonstrate the influence of bioceramic surface properties, specifically on the cells responsible for biomaterial vascularization. This made it possible to demonstrate that C166 endothelial cells are sensitive to microporosity. There are probably multiple mechanisms of action: (i) by increasing roughness; (ii) by entrapping calcium from the culture medium, creating an artificial situation of calcium depletion; and (iii) by potentially increasing the protein absorption from the culture medium, microporosity affects the availability of angiogenic factors contained in the close environment of the cells. These findings are very important for the understanding and the rationalization of research on the design of bioceramics in order to increase their biological properties. They underline the importance of understanding the phenomenon occurring at the interface between cells and biomaterials, and show that each chemical or physical parameter is sensed by cells and may interfere with the expected result.

This study also highlights an important bias of 2D in vitro static experimentation that must be taken of account in the design of future experiments. Thus, the 3D dynamic culture conditions we are currently implementing will be highly informative regarding the conclusions raised here.

## Figures and Tables

**Figure 1 jfb-14-00460-f001:**
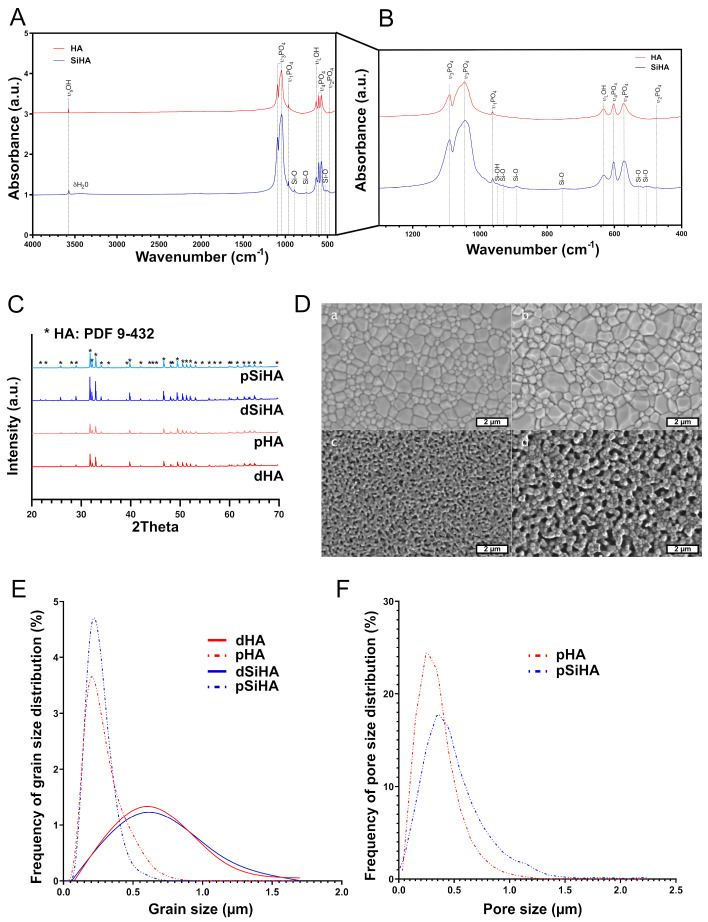
Chemical–physical characterization of HA-based ceramics. (**A**) FTIR analysis of the HA powders. (**B**) Inset of FTIR spectra of the HA powders from 400 to 1250 cm^−1^. (**C**) XRD diffractograms of the sintered ceramic pellets indexed according to ICDD-PDF card n°9-432 (*). Microstructure of HA-based ceramics. (**D**) Scanning electron microscopy micrograph of the surface of the ceramic pellets: (**a**). dense HA; (**b**). dense SiHA; (**c**). porous HA; and (**d**). porous SiHA. Scale bar (white): 2 µm. (**E**) Distribution frequency of the grain size. (**F**) Distribution frequency of the pore size.

**Figure 2 jfb-14-00460-f002:**
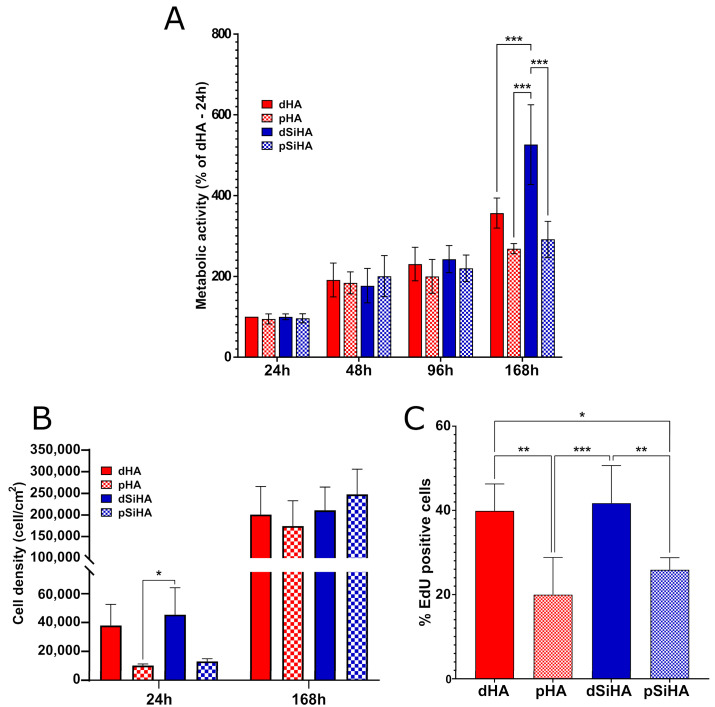
Evaluation of biocompatibility of HA ceramics towards C166 endothelial cells. (**A**) Metabolic activity evaluated by resazurin assay. (**B**) Cell density after 24 h and 168 h on HA-based ceramics, on the basis of nuclei count after pellet imaging. (**C**) Cell proliferation evaluated by EdU incorporation. Statistical analysis: one-way ANOVA followed by Tukey’s post hoc test. *: *p* ≤ 0.05; **: *p* ≤ 0.005; ***: *p* ≤ 0.001. n = 3.

**Figure 3 jfb-14-00460-f003:**
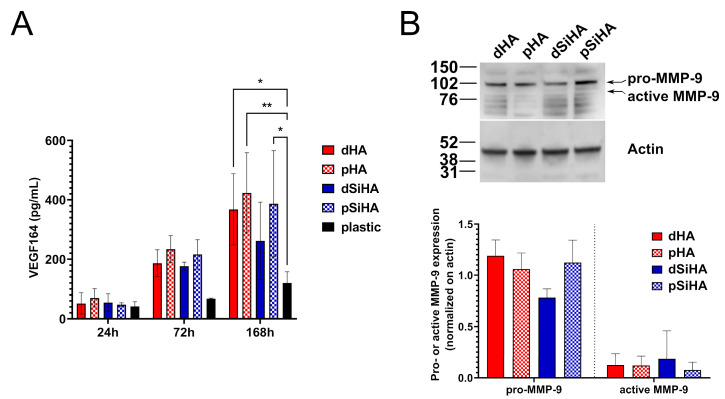
Expression of angiogenesis-related markers. (**A**) Dosage of VEGF-A secreted by C166 cells cultured at the surface of HA-based ceramics or plastic culture wells evaluated by ELISA. n = 3 independent experiments. Statistical analysis: two-way ANOVA followed by Tukey’s post hoc test. *: *p* ≤ 0.05; **: *p* ≤ 0.005. (**B**) Evaluation of expression of MMP-9 in C166 cells cultured at the surface of HA-based ceramic pellets by western-blot. Upper panel: representative MMP-9-related and actin-related signals. Lower panel: quantification by densitometry of pro- and active MMP-9 signals normalized on the signal detected for actin as housekeeping protein. N = 3 independent experiments. Statistical analysis: two-way ANOVA followed by Tukey’s post hoc test. No significant differences were detected between any of the conditions (*p* > 0.05).

**Figure 4 jfb-14-00460-f004:**
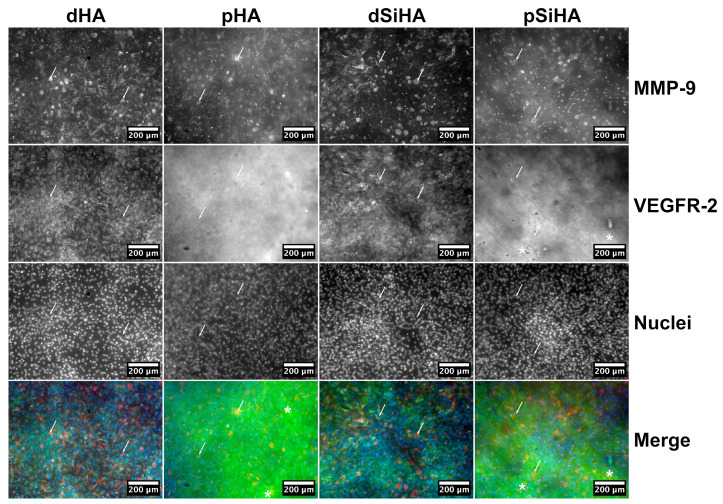
In situ expression of MMP-9 (red) and VEGFR-2 (green) as angiogenesis-related markers in C166 endothelial cells cultured on HA-based ceramic pellets after immunofluorescence staining. Nuclei were stained with Hoechst 33342 (blue). Scale bar: 200 µm. Arrows point out example of cells where the intracytoplasmic staining of MMP-9 is particularly visible. Asterisks show zones with high background. For the same figure with individual staining depicted in color (addition of false colors by post-processing), see Appendix A.

**Figure 5 jfb-14-00460-f005:**
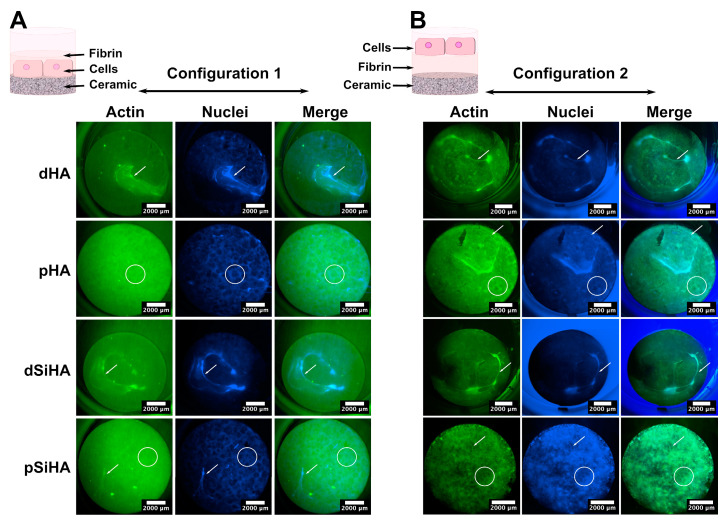
Wide-field confocal macroscopy images of C166 cell layer organization on whole ceramic pellets after tubulogenesis assay using a fibrin gel and imaged with a widefield confocal macroscope. (**A**) Configuration 1: Cells were seeded directly onto the ceramic surface and the fibrin gel poured on top. (**B**) Configuration 2: Cells were seeded on the top of the fibrin gel directly poured on the ceramic surface. Green: Actin stained with phalloidin conjugated to DyLight 488. Blue: Nuclei stained with Hoechst 33432. Scale bar: 2000 µm. Arrows point to tubule-like structures. Circles delimit examples of alveolar pattern of the cell tissue.

**Figure 6 jfb-14-00460-f006:**
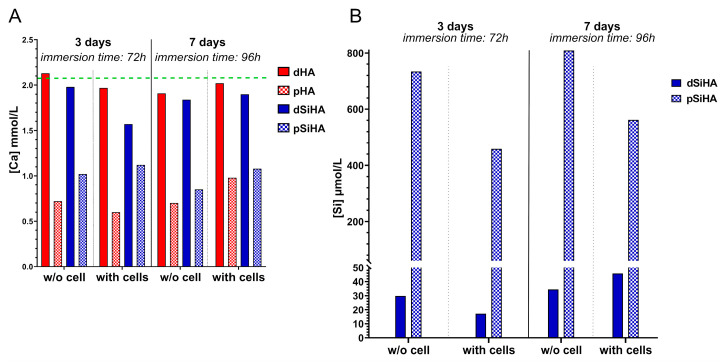
Release of chemical elements (Ca and Si) from HA-based ceramics in the C166 cell culture medium with or without cells, dosed by ICP-OES. The ceramics were immersed for 7 days in the culture medium with a total replacement of the culture medium after 3 days. (**A**) Calcium concentration. The green line materializes the concentration of calcium measured in the complete culture medium alone. (**B**) Cumulative silicon release with time. The measure was performed on pooled samples from 3 independent experiments.

**Figure 7 jfb-14-00460-f007:**
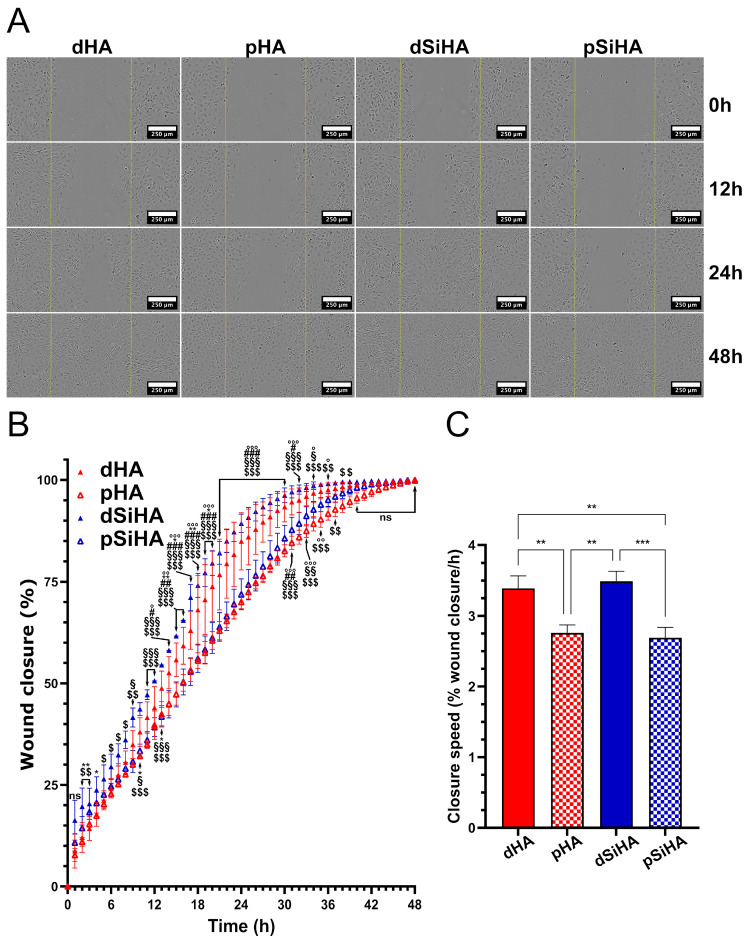
Wound healing assay performed during 48 h on C166 cell layer cultured in culture medium containing dissolution products of HA-based ceramics. (**A**) Images taken during WHA experiment. Scale bar: 250 µm. (**B**) Quantification of wound closure with time by image analysis (for details, see Section 2.2.8). Statistical analysis: two-way ANOVA followed by Tukey’s post hoc test. ns: non-significant, *p* > 0.05. one symbol: *p* ≤ 0.05, two symbols: *p* ≤ 0.005, three symbols: *p* ≤ 0.001. ◦: dHA vs. pHA, *: dHA vs. dSiHA, #: dHA vs. pSiHA, $: dSiHA vs. pHA, §: dSiHA vs. pSiHA. (**C**) Wound closure speed, calculated during the first 24 h. Statistical analysis: one-way ANOVA followed by Tukey’s post hoc test. **: *p* ≤ 0.005, ***: *p* ≤ 0.001. n = 3.

**Figure 8 jfb-14-00460-f008:**
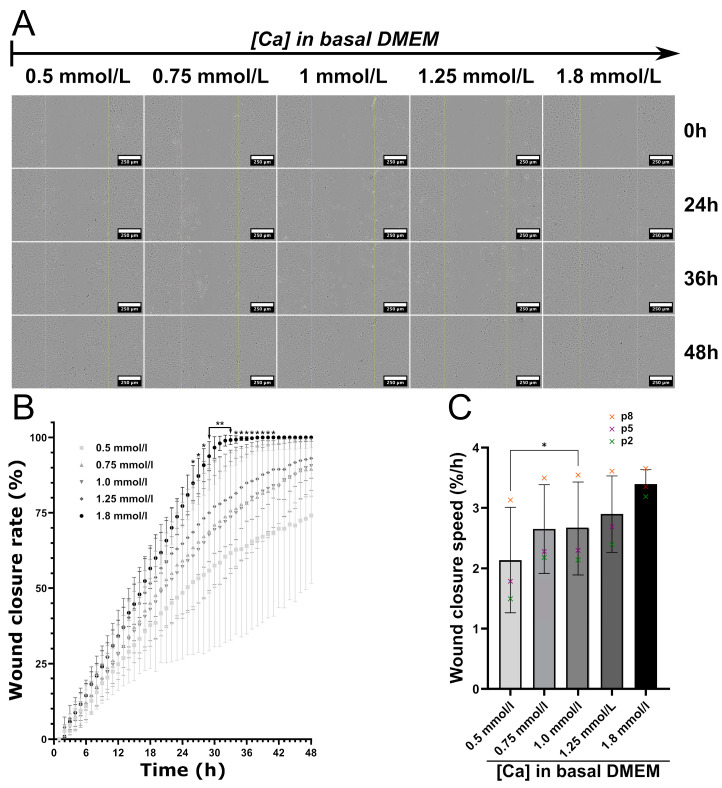
Wound healing assay performed for 48 h on C166 cell layers cultured in complete culture medium with various concentrations of calcium. (**A**) Images taken during WHA experiment. Scale bar: 250 µm. (**B**) Quantification of the wound closure with time by image analysis (for details, see Section 2.2.8). Statistical analysis: two-way ANOVA followed by Tukey’s post hoc test. *: 1.8 mmol/L vs. 0.5 mmol/L, *p* ≤ 0.05; **: 1.8 mmol/L vs. 0.5 mmol/L, *p* ≤ 0.005. (**C**) Wound closure speed, calculated during the first 24 h. Statistical analysis: one-way ANOVA followed by Tukey’s post hoc test. *: *p* ≤ 0.05. n = 3.

## Data Availability

All of the data presented in this work are available and accessible by contacting the corresponding author.

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
