# Peer review of "Microporous Hydroxyapatite-Based Ceramics Alter the Physiology of Endothelial Cells through Physical and Chemical Cues"

_jfb, 2023, doi:10.3390/jfb14090460_

Round 1
Reviewer 1 Report
Looking at the SEM images, the term of porous HA is doubted. It seems that the sintering resulted in the particle rearrangement so that the surface deterioration is occurred. The black area that the authors claimed as porous may be a void resulted from the overlap of the particles/grains. Higher SEM magnification is required to confirm the presence of porous HA. It is suggested to use the “microstructure HA” instead of “microporous HA”. The authors are suggested to compare their SEM result with the other SEMs of porous Has (examples: https://doi.org/10.1016/S0142-9612(00)00030-2, https://doi.org/10.1098/rsif.2008.0425.focus)
Authors are suggested to carefully look at the typing. Lots of typing errors are observed. Particularly the missing space between the number and unit.
How do you confirm that you have been successfully synthesized the HA and SiHA? Which analyses results that can be referred to?
The diSiHA was obtained after sintering at 1240 °C. How do you confirm that the decomposition did not occur? Previous study found that above 1200 °C, the SiHA was decomposed (See: https://doi.org/10.1002/mawe.200600007)
L86. How do you confirm that your HA and SiHA have reached the surface area 30 m2/g?
L251. What do you mean by DRX? Is it typo?
L252. What do you mean by PDF card n°9-43?
Figure 1A. The spectra were obtained from an overlaying therefore, the y-axis shall have no value.
Figure 1B. Can you explain the disappearance of u4 PO4 at SiHA sample?
L453. “The silicon quantity released by the porous SiHA ceramics was at least 25 times greater than those released by dense ceramics without cells.” à please provide the discussion on the effect of silicon release into body liquid, in term of toxicity.
Needs to be improved.
Author Response
Dear Reviewer,
Thank you for having reviewed our manuscript. Please find here after the answers to your comments (in italic):
Looking at the SEM images, the term of porous HA is doubted. It seems that the sintering resulted in the particle rearrangement so that the surface deterioration is occurred. The black area that the authors claimed as porous may be a void resulted from the overlap of the particles/grains. Higher SEM magnification is required to confirm the presence of porous HA. It is suggested to use the “microstructure HA” instead of “microporous HA”. The authors are suggested to compare their SEM result with the other SEMs of porous Has (examples: https://doi.org/10.1016/S0142-9612(00)00030-2, https://doi.org/10.1098/rsif.2008.0425.focus)
We do not understand this comment. Such images are typical and common features of porous ceramics. The microporosity (i.e. pores with a diameter in the range 1-10 µm) obtained for the ceramics called pHA and pSiHA in the manuscript resulted from an incomplete sintering well described in the literature (Champion, 2013, Acta Biomaterialia, https://doi.org/10.1016/j.actbio.2012.11.029). We published previously several works using such porous hydroxyapatite based ceramics (Lasgorceix et al., 2016, J Eur Cer Soc, https://doi.org/10.1016/j.jeurceramsoc.2015.11.020; Germaini et al., 2017; Biomed mater, https://doi.org/10.1088/1748-605X/aa69c3; Rüdrich et al, 2019, Mater Sci Eng C, https://doi.org/10.1016/j.msec.2018.12.046).
It is not possible to compare the porosities described in both papers suggested by the reviewer with the microporosity of our HA-based ceramics. Indeed, the pores in these two papers are meso- and macropores, not micropores as in our study (size 1-10 µm). The meso- and macropores described in the paper by Chang et al.,(between 100 and 340 µm) and discussed in the review by Yoshikawa et al. (10-80 µm) were produced by specific well known processes to develop this kind of porosity (incorporation of sacrificial polymer structures (beads and/or sheets) into ceramics slurry before burning and sintering to obtain macroporous ceramics with dense walls.
Please find enclosed the SEM pictures at a higher magnification.
Authors are suggested to carefully look at the typing. Lots of typing errors are observed. Particularly the missing space between the number and unit.
Typing mistakes were corrected.
How do you confirm that you have been successfully synthesized the HA and SiHA? Which analyses results that can be referred to?
Please read carefully the first paragraph of the result part which describes specifically the characterization of the ceramics used in the study (§ 3.1. Characterization of hydroxyapatite based ceramics). XRD allowed to highlight that no other crystalline phase than HA was found in the ceramics. Complementary FTIR results did not show the presence of any vibration band belonging to a chemical group of another amorphous or crystalline phase than HA or Si-HA.
The diSiHA was obtained after sintering at 1240 °C. How do you confirm that the decomposition did not occur? Previous study found that above 1200 °C, the SiHA was decomposed (See: https://doi.org/10.1002/mawe.200600007)
The XRD and FTIR results did not show another phase that could result from a decomposition of SiHA (alpha-TCP, dicalcium silicate, silicocarnotite) as previously reported in: Marchat et al., 2013, Acta Biomater. (https://doi.org/10.1016/j.actbio.2013.03.011) and in our previous works Palard et al., 2008, Acta Biomater. (https://doi.org/10.1016/j.actbio.2008.10.016). The absence of decomposition for the SiHA used in the present work was previously in Palard et al., 2008, Acta Biomater. (https://doi.org/10.1016/j.actbio.2008.10.016) in which we investigated the sintering of SiHA and demonstrated that the decomposition temperature depends of the silicon content of the initial Si-HA.
L86. How do you confirm that your HA and SiHA have reached the surface area 30 m2/g?
The BET method is routinely used to determine the surface of a powder or a ceramic part. For the powders, the specific surface area was measured by the eight-point Brunauer–Emmett–Teller method (Micromeritics ASAP 2000) using adsorption N2 (for pHA and pSiHA) after degassing powders under vacuum at 200 °C for 1 h.
L251. What do you mean by DRX? Is it typo?
Thank you. DRX is a typo, the right word is XRD (X-Ray Diffraction). It was corrected.
L252. What do you mean by PDF card n°9-43?
It corresponds to the Powder Diffraction file number (9-432) of HA from ICCD (International Centre for Diffraction Data): https://www.icdd.com/. This information was added in the section materials ans methods.
Figure 1A. The spectra were obtained from an overlaying therefore, the y-axis shall have no value.
You are right, it is why the absorbance is followed by a.u. for arbitrary units.
Figure 1B. Can you explain the disappearance of u4 PO4 at SiHA sample?
We do not understand the comment, as it can be seen in Figures 1A and 1B, u4 PO4 is present for SiHA (blue line). It is important to note that silicon substitutes as silicate SiO4 for PO4 groups in HA lattice resulting in the following formula for SiHA (molar substitution rate of Si=0.4 mol): Ca10(PO4)5.6(SiHO4)0.4(OH)1.6 (Palard et al., 2008). This small substitution ratio and decrease of PO4 groups does not change significantly the intensity of the PO4 bands.
L453. “The silicon quantity released by the porous SiHA ceramics was at least 25 times greater than those released by dense ceramics without cells.” please provide the discussion on the effect of silicon release into body liquid, in term of toxicity.
A paragraph about silicon toxicity was added in the discussion part: “Microporosity was shown to increase bone cell adhesion through this phenomenon [19]. Microporosity also enhanced the release of silicon in the culture medium, reaching around 800 mmol/l (22.73 mg/l) after 7 days without cells. The dosage method employed here did not allow to identify the chemical form of the silicon released from Si-doped HA ceramics. Silicon is incorporated as silicate ion, so it should be released as silicate ions or as metabolizable silicic acid [60] as it was modeled in a recent study by Chappell et al. [60]. Silicon is bioavailable as orthosilicic acid and has a positive role on bone metabolism [61] and bone mineralization [60] when its absence is deleterious for bone homeostasis. The daily dietary silicon intake is estimated from 20 to 50 mg [60,62]. No toxicity of dietary silicon was detected above 50,000 mg in rats [63]. Silicon is tightly regulated in the organ-ism, and cell specific transporters were recently identified [64,65]. It was shown that silicic acid produced by the degradation of silicon quantum dots is eliminated through urine [66]. In the context of implantation of silicon-containing ceramics as bone graft substitutes a potential toxicity due to silicon can be reasonably ruled out especially as micro-macro-porous silicon substituted calcium phosphate scaffolds are currently used in clinic [12,67,68].”
Best regards,
Amandine Magnaudeix

Reviewer 2 Report
The authors developed a microporous hydroxyapatite-based ceramics scaffold varying either by their chemical composition (pure versus silicon-doped HA) or by their microstructure (dense versus microporous ceramics) to alter the physiology of endothelial cells through physical and chemical cues . The results are of great interest to novel technologies and biomaterials research communities. The reviewer suggests accepting this manuscript after taking care of the following issues.
Specific comments:
- Overall a very interesting article with high value for researchers as well as for possible future exploitation in clinical application.
- A very impressive list of methods was used in the study, hence the analysis is very thorough and the claims therefore mostly experimentally sound.
- The study is quite comprehensive and has achieved very good results. Authors should write the abstract section in more detail.
- The ideas involved in the work should be explained fully in the introduction part itself. Introduction needs to be improved and some recent works could be cited.
- This article lacks experimental verification in vivo, which are more convincing evidence.
- In Figure 7, the drawing of the wound healing assay looks too casual. Please redesign them!
- Provide recent references not old references according to journal format and also renumber them properly.
- There are some formatting mistakes in the references section, I suggest the authors check and correct them. For example, there are incomplete references or with erroneous data, others with typos in the journal name or chemical formulae in the title. For example, this issue can be seen in ref. no. 26
After a careful review of the manuscript, I am happy that I can recommend this article for publication after a minor revision with all the above suggestions.
Author Response
Dear Reviewer,
Thank you for having reviewed our manuscript. Please find here after the answers to your comments (in italic):
The authors developed a microporous hydroxyapatite-based ceramics scaffold varying either by their chemical composition (pure versus silicon-doped HA) or by their microstructure (dense versus microporous ceramics) to alter the physiology of endothelial cells through physical and chemical cues. The results are of great interest to novel technologies and biomaterials research communities. The reviewer suggests accepting this manuscript after taking care of the following issues.
Specific comments:
Overall a very interesting article with high value for researchers as well as for possible future exploitation in clinical application.
A very impressive list of methods was used in the study, hence the analysis is very thorough and the claims therefore mostly experimentally sound.
The study is quite comprehensive and has achieved very good results. Authors should write the abstract section in more detail.
Thank you for your constructive comments. The abstract was rewritten as suggested.
The ideas involved in the work should be explained fully in the introduction part itself. Introduction needs to be improved and some recent works could be cited.
We improved the introduction section, corrected and add references of recent works.
This article lacks experimental verification in vivo, which are more convincing evidence.
Thank you for your comment. The concepts and the results presented in the submitted work have important implications for in vitro studies. In vitro results are required to conduct further in vivo experiments respecting ethical local rules.
In Figure 7, the drawing of the wound healing assay looks too casual. Please redesign them!
The figure 7 was redesigned, the drawing was replaced by an arrow.
Provide recent references not old references according to journal format and also renumber them properly.
The references and their numbering were checked. New references to more recent works were added when available and relevant with the text.
There are some formatting mistakes in the references section, I suggest the authors check and correct them. For example, there are incomplete references or with erroneous data, others with typos in the journal name or chemical formulae in the title. For example, this issue can be seen in ref. no. 26
The references were checked and corrected.
After a careful review of the manuscript, I am happy that I can recommend this article for publication after a minor revision with all the above suggestions.
Best regards,
Amandine Magnaudeix
Reviewer 3 Report
This study explored the micro-structure and particular ions addition on the CPC manufacture influence the C166 EC behavior, used to speculate and interpret the angiogenic efficacy in bone regeneration. This is an essential and helpful study. Here, some points need to be revised and delineated more clearly. The authors need thoroughly and carefully check the parameters and numbers in the manuscript.
1) In the Abstract, please briefly delineate the current study's motivation, problems or hypothesis, and unmet needs. And, please give the exact value and statistical data and trace the phenomenon of significant results, rather than only showing “… ability to assemble into tubule-like structures.”
2) The manufacturing protocol and parameters of ceramics preparation need to be finetuned more clearly. The oxygen, pressure, temp, oven time, weight ratio, mold,…etc. should be delineated.
3) The figure caption has mistakes.
4) If the figure has images, please describe and show the scale bar correctly.
5) The authors showed *, **, and *** to represent the statistical value. However, please carefully check the label and corresponding caption.
6) In Figure 4, please label the signals on the images that the authors would like to emphasize with stars or arrows.
7) In Figure 4, the images are immunohistochemical staining or immunofluorescent staining; please correctly delineate the experiments
8) Again, please use markers to represent the interested area in Figure 5 and briefly delineate the experimental and examining methods.
9) In Figure 6, please briefly delineate the experimental and examining methods, especially the quantification methods.
10) The authors must describe the rationale for using the C166 cell rather than other osteoblastic cell lineages or progenitor cells.
11) The reviewers hope the authors explain why the study does not use positive control, commercialized products, well-developed scaffolds, or function, well-known products for comparison in parallel. Although you can find the authors, they compared many concentrations, time, dHA, pHA, dSiHA, pSiHA, with and without cells,…etc. However, throughout the whole study, a lack of sophisticated positive control for experimental comparison.
12) Finally, the conclusion should be revised thoroughly. Please, the authors show the most critical summaries, data and interpretation, scientific findings, novelties, and future applications.
Check some mistakes in the figure captions.
Author Response
Dear reviewer,
Thank you for having reviewed our manuscript and for your valuable comments.
You will find hereafter our answers to yours questions and comments (in italic):
This study explored the micro-structure and particular ions addition on the CPC manufacture influence the C166 EC behavior, used to speculate and interpret the angiogenic efficacy in bone regeneration. This is an essential and helpful study. Here, some points need to be revised and delineated more clearly. The authors need thoroughly and carefully check the parameters and numbers in the manuscript.
1) In the Abstract, please briefly delineate the current study's motivation, problems or hypothesis, and unmet needs. And, please give the exact value and statistical data and trace the phenomenon of significant results, rather than only showing “… ability to assemble into tubule-like structures.”
The abstract was rewritten accordingly within the word limit imposed by the journal format (200 words).
2) The manufacturing protocol and parameters of ceramics preparation need to be finetuned more clearly. The oxygen, pressure, temp, oven time, weight ratio, mold,…etc. should be delineated.
Powder synthesis was performed under argon atmosphere as explained in the joined references since this procedure is routinely used in the lab and already published and the Material and method section is quite long. The mold is described in the text: it is a Specac die for uniaxial pressing.
Please see § 2.1. Ceramic processing and characterization, the asked information are highlighted:
“HA (Ca10(PO4)6(OH)2) and SiHA (Si molar ratio: 0.4; Ca10(PO4)5.6(SiHO4)0.4(OH)1.6) powders were synthetized by aqueous precipitation. Detailed protocols can be found in previous works [17,20]. The hence obtained raw powders were calcined at 650°C during 30 min for HA and 700°C during 2 h for SiHA, under air (LH30/13, Nabertherm) to reach a specific surface area around 30 m2/g which facilitated the compaction of powders. Specific surface area of powders was checked according to the eight-point Brunauer–Emmett–Teller method (BET - Micromeritics ASAP 2000) using adsorption of N2 after degassing under vacuum at 200°C for 1 h. Afterwards, pellets were shaped by uniaxial pressing at 125 MPa in a 10 mm diameter cylindrical die (Specac).
The sintering cycles were adjusted to obtain a similar microstructure for dense HA and SiHA ceramics (referred as dHA and dSiHA) on the one hand, and for porous ones (referred as pHA and pSiHA) on the other hand. The sintering parameters were 1200°C for 30 min for dHA and 1240°C for 30 min for dSiHA (LHT04/17 furnace, Nabertherm) and 1000°C for 30 min and 1160°C for 30 min (LH30/13 furnace, Nabertherm) for pHA and pSiHA, respectively. Sintering was performed under air.”
3) The figure caption has mistakes.
The figure captions were checked and corrected.
4) If the figure has images, please describe and show the scale bar correctly.
The scale bars were increased in size for figure 1. For immunofluorescence microscopy images, scale bars were incrusted with ImageJ, scale bar is described in every caption.
5) The authors showed *, **, and *** to represent the statistical value. However, please carefully check the label and corresponding caption.
The captions were corrected. There was a mistake with a duplication of two versions of the figure 2 instead of figure 2 and 3 (ELISA dosage of VEGF-A and evaluation of the MMP-9 protein expression by western-blotting). This was corrected. All the reference to statistics-related symbols were checked.
6) In Figure 4, please label the signals on the images that the authors would like to emphasize with stars or arrows.
Arrows and stars were added to help the reading of the figure 4. A supplementary figure (Figure S2) was added. It corresponds to figure 4 in which black and white images were colorized to help reading.
7) In Figure 4, the images are immunohistochemical staining or immunofluorescent staining; please correctly delineate the experiments
The figure caption was corrected accordingly.
8) Again, please use markers to represent the interested area in Figure 5 and briefly delineate the experimental and examining methods.
The Figure 5 caption was rewritten. Arrows and circles were added to help readers, their meaning is explained in the caption.
9) In Figure 6, please briefly delineate the experimental and examining methods, especially the quantification methods.
The Figure 6 caption was rewritten. The quantification method was given in the material and methods part.
10) The authors must describe the rationale for using the C166 cell rather than other osteoblastic cell lineages or progenitor cells.
C166 are not osteoblastic cell line nor osteoblastic progenitor because the main objective of this work was to study the impact of ceramic physical-chemical properties on vascularizing cells, i.e. endothelial cells. Vascularization is of prime importance for new tissue ingrowth in bone scaffolds but remains very few investigated in vitro. We hope that the revised introduction of the manuscript clarifies this point. On the opposite numerous data were already published in the literature regarding the influence of silicon or that of microporosity on osteoblastic cells, including from our research team.
11) The reviewers hope the authors explain why the study does not use positive control, commercialized products, well-developed scaffolds, or function, well-known products for comparison in parallel. Although you can find the authors, they compared many concentrations, time, dHA, pHA, dSiHA, pSiHA, with and without cells,…etc. However, throughout the whole study, a lack of sophisticated positive control for experimental comparison.
In our study, dense HA (dHA) is a positive control because stochiometric dense HA is clinically used since more than 40 years. We did not use 3D scaffold in order to not increase the number of parameters and to be able to compare same physical or chemical features two-by-two. A work in progress in our laboratory is devoted to study these parameters applied to scaffolds shaped by stereolithography (Danty et al., Solid State Phenomena, 2022, doi: 10.4028/p-zn71xt)
12) Finally, the conclusion should be revised thoroughly. Please, the authors show the most critical summaries, data and interpretation, scientific findings, novelties, and future applications.
The conclusion was rewritten according to your suggestions.
Best regards,
Amandine Magnaudeix
Reviewer 4 Report
This manuscript is devoted to the current direction of modern scaffold vascularization. The results obtained are described in detail in the manuscript and comparative studies with the results of other authors are carried out. It is recommended to improve the quality of some drawings. This manuscript is recommended for publication in the journal.
It is necessary to provide the original color photos to Figure 4 in order to evaluate Merge.
Author Response
Dear reviewer,
Thank you for having reviewed our manuscripts anf for your questions and comments.
You will find hereafter our answers (in italic) :
This manuscript is devoted to the current direction of modern scaffold vascularization. The results obtained are described in detail in the manuscript and comparative studies with the results of other authors are carried out. It is recommended to improve the quality of some drawings. This manuscript is recommended for publication in the journal.
It is necessary to provide the original color photos to Figure 4 in order to evaluate Merge.
Thank you for the careful reading of the manuscript and your suggestions. The quality of figures was improved.
The photos were taken with a CCD camera so they were natively in black and white. We provided the photos with assigned false colors in supplementary data (supplementary figure S2).
Best regards,
Amandine Magnaudeix
Round 2
Reviewer 2 Report
The revised version could be published because the authors have re-writed well and reponded the critical concerns.